# 'She convinced me'- partner involvement in choosing a high risk birth setting against medical advice in the Netherlands: A qualitative analysis

Martine Hollander[1]*, Esteriek de Miranda[2], Anne-Marike Smit[3], Irene de Graaf[2], Frank Vandenbussche[1], Jeroen van Dillen[1], Lianne Holten[3]

**1** Department of Obstetrics, Radboud University Medical Center, Amalia Children's Hospital, Nijmegen, the Netherlands, **2** Academic Medical Center, Department of Obstetrics, Amsterdam UMC, Amsterdam, the Netherlands, **3** AVAG School of Midwifery and Amsterdam UMC, VU/EMGO Research Institute, Amsterdam, the Netherlands

* martine.hollander@radboudumc.nl

**Data Availability Statement:** The data for this study consist of transcripts of interviews with 21 participants. All transcripts are in Dutch. These

## Abstract

Home births in high risk pregnancies and unassisted childbirth seem to be increasing in the Netherlands. There is a lack of qualitative data on women's partners' involvement in these choices in the Dutch maternity care system, where integrated midwifery care and home birth are regular options in low risk pregnancies. The majority of available literature focuses on the women's motivations, while the partner's influence on these decisions is much less well understood. We aimed to examine partners' involvement in the decision to birth outside the system, in order to provide medical professionals with insight and recommendations regarding their interactions with these partners in the outpatient clinic. An exploratory qualitative research design with a constructivist approach and a grounded theory method were used. In-depth interviews were performed with twenty-one partners on their involvement in the decision to go against medical advice in choosing a high risk childbirth setting. Open, axial and selective coding of the interview data was done in order to generate themes. Four main themes were found: 1) Talking it through, 2) A shared vision, 3) Defending our views, and 4) Doing it together. One overarching theme emerged that covered all other themes: 'She convinced me'. These data show that the idea to choose a high risk birth setting almost invariably originated with the women, who did most of the research online, filtered the information and convinced the partners of the merit of their plans. Once the partners were convinced, they took a very active and supportive role in defending the plan to the outside world, as well as in preparing for the birth. Maternity care providers can use these findings in cases where there is a discrepancy between the wishes of the woman and the advice of the professional, so they can attempt to involve partners actively during consultations in pregnancy. That will ensure that partners also receive information on all options, risks and benefits of possible birth choices, and that they are truly in support of a final plan.

contain identifying items and are therefore sensitive to privacy issues. Participants only allowed the interviews under promise of anonimity. The authors are expressly forbidden by the participants to make the full content of the interviews public. Therefore, the authors are ethically bound not to publish the full content, as publishing an entire interview would be easily traced back to the participant in question. Anonymized excerpts from the full transcripts can be made available to qualified researchers by the medical ethical committee of the Radboud University, who can be contacted at commissiemensgebondenonderzoek@radboudumc.nl.

**Funding:** MH's travel expenses to conduct the interviews were re-embursed by her employer, the department of obstetrics at the Radboud University Medical Center, Nijmegen, the Netherlands. LH's travel expenses to conduct the interviews were re-embursed by her employer, AVAG school of midwifery, Amsterdam, the Netherlands. No other funding was obtained.

**Competing interests:** The authors have declared that no competing interests exist.

## Introduction

Since approximately the 1960's, men have been increasingly involved in the process of labor and birth, and are, in most western high income countries, generally expected to be present in the labor room[1]. However, many men still feel somewhat disassociated from the process: the majority don't attend most antenatal appointments and feel that maternity care providers don't really include them when they counsel their pregnant partners[2–5]. Because of this, many expectant fathers experience a lack of information, which they feel is a barrier to their involvement in the process of decision making concerning childbirth[6–14].

To date, there are limited data on partner involvement in decisions concerning place of birth and choice of birth attendant, neither when these choices conformed to the local maternity care system, nor when they were against medical advice.

In most high income countries, hospital birth has become the norm, and home birth, even in low risk pregnancies, is considered by mainstream maternity care providers as against medical advice[15]. However, the Netherlands, and, to a lesser extent, New Zealand, Canada and the United Kingdom, have a system in place where low risk pregnant women may opt for a home birth with a (community) midwife. Women with a high risk pregnancy are advised to go to a hospital to give birth under supervision of a gynecologist.

In the Netherlands, Hendrix et al.[16] investigated whether fathers participated in decision-making concerning the choice for home birth versus hospital birth in low risk women, and found that 60% of fathers reported that they were involved in the decision. In a recent study from the UK[17], where home birth in a low risk pregnancy is considered an acceptable choice, 21 male partners of low risk pregnant women were interviewed regarding their choice for place of birth. All partners stated that their choice for hospital was an automatic one, and that they would have been very unhappy if their wives had suggested the idea of home birth. This finding is in accordance with an Irish study by Sweeney et al.[18], who interviewed eight male partners whose wives opted for a home birth. All partners initially resisted the idea, even though it was (although an uncommon choice) not against medical advice.

Several studies have been done in countries where all home births are against medical advice. Three Scandinavian studies[19–21] and a Spanish study[22] have similar findings as the studies quoted above: the idea for a home birth came from the women. The men had doubts at first, but were eventually in agreement with the women. However, no studies have been done as yet in countries with integrated home birth for low risk women concerning partner involvement in the decision to go against medical advice and have a home birth in a high risk pregnancy or an unassisted childbirth (UC).

In the Netherlands, 30% of all births are attended by community midwives, about half of which are home births[23]. Almost all of these are low risk births, however, a small group of women chooses to have a home birth in a high risk pregnancy, attended by a community midwife. In addition, another small group opts for a UC, which is their legal right. Both choices are explicitly against medical advice. In 2017, we published a qualitative study among 28 women who made such a choice, examining their motivations for doing so[24]. The themes that emerged from these interviews centered around dissatisfaction with the regular system of maternity care, trust in nature and their own capacity for giving birth, conflict with maternity care professionals and a search for alternative care. An overarching theme of "fear" (of unnecessary interventions, loss of autonomy, and of provider's fear of legal consequences) was found.

This current study set out to examine the involvement of Dutch partners in the decision-making process concerning a home birth in a high risk pregnancy or UC in a country in which home birth for low risk women is integrated in regular maternity care. This information could

help health care providers to involve partners in conversations concerning decision making surrounding choice of place of birth, whether within or outside the system. To that end, we interviewed the majority of the partners of the women who were the subject of our previous study.

## Methods

The COREQ criteria for reporting qualitative research[25] were used to ensure a complete and correct approach to data collection and analysis. Permission for this study was sought from ethical committees of both the University of Amsterdam and the Radboud University in Nijmegen, the Netherlands. Both committees considered this study as not requiring ethical permission.

### Research team and reflexivity

The majority of interviews (18) were conducted by the first and last authors (MH and LH), while the remaining five were conducted by three midwifery students under the supervision of the last author (LH). All interviewers are women, and both MH and LH are experienced in conducting interviews for qualitative studies. All interviewers have a background in midwifery/obstetrics, and LH also has a background in anthropology. EdM participated in designing the study, JvD, IdG, AS, FV and EdM gave constructive criticism on earlier drafts of this article, and approved the final version.

None of the partners were known to the interviewers prior to the interviews. The partners were aware that the interviewers had a professional interest in birth outside guidelines, women's rights and the Dutch maternity care system, and were known in the field as supporters of women's rights and autonomy. All participants gave verbal informed consent for their quotes to be used.

### Study design

This exploratory qualitative research used a constructivist approach and a grounded theory method[26]. This study is part of a larger project exploring out of the system birth, the Wonderstudy[27], in which we also interviewed women who went against medical advice in their birth choices (home birth in a high risk pregnancy or UC), their partners and their midwives. All partners were contacted through the women. The interviews with the women were the subject of previously published research by this group[24]. All partners who were asked for an interview agreed to be interviewed. Interviews were conducted between the summer of 2014 and the winter of 2016, and two partners were interviewed twice. Interviews concerned births that took place between 2010 and 2015. The majority of interviews took place in the partners' home, with one exception, which took place in a public place. Most often the partner was interviewed alone, although in several cases, the woman was present in the room and occasionally joined in the conversation. Location of the interview and presence of the woman was by partner's choice. All partners gave consent for their quotes to be used in this article.

Interviews were semi-structured by use of a topic list (Fig 1) but were allowed to flow freely, lasting between 30 and 75 minutes. The topic list was based on themes from the literature on women's motivations[28–32], and topics were added as new (sub-)themes if mentioned in other interviews. A digital sound recorder was used and data were then transcribed verbatim by either a commercial company or volunteer medical students. All data were stored anonymously in a password protected university database.

Where did your partner intend to give birth? Alone or with a midwife?

Did you two have conversations about this? Who brought it up first? What were your thoughts?

Which factors do you believe influenced the decision to go against medical advice?

Did you see an increase in risk? Did you discuss this together? Did you make emergency plans?

What do you think a partner's role should be during the birth? How did you prepare?

Did you speak to others about your decision? What did they say?

How did the birth go? How was it for you, looking back now? Would you make the same choice again?

What is a good birth? What do you think of the current maternity care system? What should we change?

**Fig 1. Topic list.** List of topics used during the interviews.

### Data analysis

All interviews were coded by the first author (MH). Data analysis was done using qualitative data analysis program MaxQDA (VERBI GmbH™). After 10 interviews, the last author (LH) coded an interview as a peer review to see if any codes were missing, and one code was added. Coding was started from the bottom up, with each interview adding and building on the coding tree (Fig 2). Codes were then grouped into themes and subthemes. Data saturation was reached after 18 interviews. Consensus on the final coding tree was reached through discussion between MH and LH after all coding was completed. All quotes were translated from Dutch into English by MH.

### Results

Twenty-one male partners were interviewed involving 27 births, the majority of which were UC's (7), home Vaginal Births after Cesarean (VBAC)'s (6), home breech births (4) and twin home births (2). There were 25 live births and two intra partum deaths. The intra partum deaths involved one case of a quick and uncomplicated term breech birth, where the baby was born without vital signs, resuscitated and died several days later. The other case was a protracted and unmonitored term VBAC, where the baby died during labor, presumably due to asphyxia. There was no ruptured uterus. This was confirmed afterwards at cesarean section. The age of the partners ranged from 24 to 43 years, with almost half (43%) being older than 35. Most partners (86%) were employed, and two thirds (62%) were highly educated, of whom five (24%) had a university degree. Two partners were of Moroccan descent, all others were ethnic Dutch. For a minority of the partners (21%), the birth relevant for the interview was their wife/girlfriend's first birth, while for most partners (67%), it was their wife/girlfriend's second and/or third birth which was the subject of the interview. Almost half (43%) of the secondary care indications concerned a planned VBAC. Demographic data can be found in Table 1.

**Talking it through**

- Previous bad experience in regular care
  - Traumatic experience
    . Partner felt she had failed as a woman
    . Working on dealing with the trauma
  - Bad communication skills
  - No flexibility
  - No continuity
  - Conflict in regular care
  - No informed consent
  - No shared decision making
    . Threats, legal action
    . Paternalism
- Dissatisfaction with the regular system
  - Things go wrong in hospitals too
  - Regular carers are afraid
    . Risk talk, exaggerating risk
        +Lack of information
  - EBM is limited
    . Interventions cause pathology, create risk
    . Carers interrupt the process
    . You need to tailor care
    . Midwife/obstetrician left us little choice/recommended hospital
    . Creating more and more conditions to be met
    . Guidelines are not laws
  - Hospitals only focus on the medical side
- She has done all/most of the research

  - Social media

- Weighing the risks
  - Deciding moment
    . We would have been willing to go to hospital
  - Letting go of fear
  - Accepting a bad outcome, dealing with a bad outcome
- But it's my baby too
  - But I prefer hospital just in case
  - But healthy mother and child are the most important

**Defending our views**

- Opinions of family and friends
  - Not discussing it
  - Spreading the word, convincing others
  - They suggested it
  - Going against medical advice is scary/difficult
    . Leading to doubts
    . They are the experts
    . People can point fingers if it goes badly
  - Fear of birth, aversion to risk
    . Difficult to explain, no understanding
    . You should follow advice
  - Favourable
- Protecting and defending her against caregivers
  - Going to the hospital in her place

**Doing it together**

- Preparations for birth
  - Getting tests
  - Designated clinic, getting all the information
    . Not discussing your plans with the hospital
    . Making sure the hospital knows our plans
  - Thinking about plan B
    . Willing to be referred if things go wrong
  - Taking a course
  - Doing research, reading up
- Writing a birth plan
  - Wanting and not getting a water birth
    . Water birth/birthing position
- Finding a midwife who agrees
  - Search for a new midwife halfway through
  - Holistic midwives are misunderstood
    . Holistic midwives should be reimbursed/more integrated
  - Travel distances, too late
  - Role of the midwife
    . To be there just in case
    . We don't need her (UC)
  - Connection with the midwife
- Men should have a larger role during pregnancy and birth

**Fig 2. Code tree.**

Home births are described as 'attempted' when the birth was not completed at home and transfer to the hospital was necessary, and, in one case, as 'intended', when an ante partum complication occurred, making it necessary for the couple to abandon their plans for a UC.

After grounded theory analysis of the 23 interviews, four main themes emerged: "Talking it through", "A shared vision", "Defending our views", and "Doing it together". After careful consideration of the data, one overarching theme emerged, which was "She convinced me".

## Talking it through: Not an easy choice

In all but one case (an attempted home VBAC), the idea to go against medical advice and choose a UC or a home birth in a high risk pregnancy was first suggested to the partners by

**Table 1. Partner characteristics (N = 21, involving 27 births).**

| Partner characteristics | N |
|---|---|
| **Age at relevant birth (years)** | |
| 20–25 | 2 |
| >25–30 | 1 |
| >30–35 | 9 |
| >35–40 | 4 |
| >40–45 | 5 |
| **Employed** | |
| Yes | 18 |
| No (still a student) | 3 |
| **Highest education** | |
| High School | 5 |
| Vocational training | 3 |
| College | 8 |
| University | 5 |
| **Ethnic origin** | |
| Moroccan | 2 |
| Dutch | 19 |
| **Marital status at time of relevant birth** | |
| Married | 18 |
| Living together | 3 |
| **Indication for secondary care** | **14** |
| VBAC (1 also diabetes type I) | 6 |
| Breech (1 also post term) | 4 |
| Twins | 2 |
| Previous postpartum hemorrhage (>1000 ml) and manual placenta removal | 1 |
| High body mass index (> 35) | 1 |
| **Unassisted childbirth (UC)** | **7** |
| **Perinatal death** | **2** |
| Breech | 1 |
| VBAC | 1 |
| **Wife/girlfriend's parity after relevant birth** | |
| 1 | 4 |
| 2 | 8 |
| 3 | 6 |
| 4 | 2 |
| 5 | 0 |
| 6 | 1 |

their wives/girlfriends, whereas the partners themselves had no strong personal feelings or preferences about the birth setting. Deciding to go against medical advice in their birth choices was not a decision that was made overnight. All partners reported that this required a substantial amount of "talking things through" in order for them to agree to the proposed plan.

Most of the partners in this study stated that the idea originated in a negative experience in regular maternity care. In either a previous or the current pregnancy, partners and their wives/ girlfriends had experienced bad communication skills, paternalism, and a lack of flexibility, continuity, informed consent and shared decision making from maternity care providers. In

many cases, there had been a conflict between the couple and providers, and in some cases even threats of legal action being taken by care providers against the couple:

> "So it was the same again: "Aren't you afraid your child will die, your wife will die?' Really putting pressure on us and then I got a bit angry like: This isn't health care, this is not thinking things through together with people, this is only making things hard for them." (partner 8, home VBAC)

Partners reported that, because of these negative experiences, their wives/girlfriends decided to look for an alternative to regular care. They stated that the women suggested the idea of going against medical advice after doing extensive research, mostly on social media, which was then presented to the partners:

> "[She] really took all this in like a sponge, and spent night after night searching on Facebook groups about this subject, and the book 'Free Birth', [. . .] and sometimes she would show me things." (partner 1, previous PPH and MPV)

> "[Wife] spent a lot of time searching the internet for all kinds of articles. I didn't have the opportunity to spend that much time on it, so. . .at one point I just assumed she knew more than I did." (partner 12, attempted twin home birth)

During the interviews, it became clear that several partners had not done much research for themselves, rather they had been given limited information on the risks that the proposed birth plan entailed, and they appeared to be not quite aware of what risks they were taking. It appeared that the information at their disposal largely originated from their wives/girlfriends, in contrast to what they might have been told by medical professionals if they had been present for most or all consultations:

> "I wonder if this was a high risk pregnancy. Is this always considered high risk? [. . .] OK, I did not experience this as high risk. [. . .] Didn't realize, either. I did think it was higher risk than a singleton pregnancy, but. . .didn't really consciously think about it." (partner 14, twin home birth)

> "What are the chances of an actual uterine rupture? They are no larger than that a firstborn comes out with a prolapsed umbilical cord. [. . .] So we felt like: what is the big deal?" (partner 20, attempted home VBAC, intra partum fetal death)

The information presented to them by the women convinced the partners that interventions could also cause pathology, and presence of care providers could interrupt the process of childbirth. They felt that hospital care was only focused on the biomedical model of childbirth, not taking into account that giving birth is a major life event. In addition, they stressed that evidence based medicine is limited, and not always applicable:

> "Giving birth is a natural thing. Basically, apart from a few complications, every woman can give birth normally. When you are in the hospital, it is a much smaller step [. . .] to have an intervention. If you are not in the hospital, you will have to do more yourself. [. . .] One intervention often leads to the next, with the whole cascade we had last time." (partner 21, UC)

The partners had the impression that maternity care providers were afraid of legal consequences in case of a bad outcome, which would cause them to exaggerate risks communicated to the couple:

*"We mostly encountered people who were very afraid. Very afraid things would go wrong."* *(partner 10, home breech birth)*

Midwives and obstetricians were accused of creating more and more conditions to be met, leaving the couples little choice but to either agree to all proposed measures or move outside the system and choose a home birth in a high risk pregnancy, or an unassisted childbirth:

*"And every time they made up another reason not to agree [to what we wanted]." (partner 18, home VBAC)*

Most couples had extensive conversations about the subject of a home birth in a high risk pregnancy, or an unassisted childbirth, in which they weighed the risks together:

*"I was often aware that it is actually just. . .actually a big risk. . .that you are at home. And you have to wait ten minutes for an ambulance. . ..and you don't have ten minutes." (partner 10, home breech birth)*

Agreeing to support their wives/girlfriends in a decision to choose a home birth in a high risk pregnancy or a UC was not an easy choice to make for some partners, and required them to let go of their fears and embrace trust in a good outcome:

*"Well, that was a difficult process for me too. Actually, everything that you encounter, that I could encounter, that I am afraid of. . ..is that real, is it justified or is there a solution to be found?" (partner 19, UC)*

After much discussion and soul-searching, partners reported that they were convinced by the women's arguments and agreed to their suggestions, which meant that they would also accept a bad outcome, if it came to that:

*"We talked through the implications together. What it could mean and what we would do with this [a bad outcome]. We clearly said to each other: we make this choice and we take responsibility for it ourselves." (partner 9, home breech birth)*

*"What does it mean that things could go wrong? Both with [wife] and with [baby]. And what does this do to me? And what do we need to go down this path together?" (partner 16, intended UC)*

In summary, a (previous) negative experience in regular maternity care led the partner's pregnant wives/girlfriends to suggest going against medical advice in their birth choices. This necessitated much discussion, during which the partners were convinced by the women's arguments about negative aspects of maternity care and their research on social media into alternative birth options, and agreed to go down this path, support their wife/girlfriend and accept responsibility for this decision and the outcome together.

## A shared vision

During discussions, the partners became convinced by the women's arguments, came to share their views on the best birth option for them, and agreed to the plan for a home birth in a high risk pregnancy, or an unassisted childbirth. This became a shared vision, in which not only the physical process of giving birth, but also the process of becoming a family played an important role:

*"I come from a family myself, where closeness is very important. That is a home situation. I want it to be like that for my children, I want to create that and contribute to it. I think that has only strengthened our choice for home."* (partner 6, UC)

Many partners believed that the way the birth went and how the baby came into the world would affect the dynamics of their family, their bond with the baby, and perhaps even the character of the child:

*"And the problem with [name child] actually was that we gave away too much [control] to those doctors, that our son did not actually have a connection with us. Not until much later. And that caused him a lot of stress. He slept badly. At home, he almost never slept. Awake a lot, crying a lot, those sort of things."* (partner 18, home VBAC)

Partners stated that they felt that birth is a natural event, which has the best chance of proceeding without problems if left alone. They were convinced that in order to have a good birth, you need to do what feels right for you, and women have to be able to relax and have faith in themselves and those around them:

*"When a woman gets into the right mood, when she withdraws almost like an animal in the bushes and does her thing and closes off and when there is quiet and she makes her own [endorphins] or whatever is necessary, then birth will just take its course."* (partner 1, previous PPH and MPV)

For some partners the ultimate form of trust is unassisted childbirth:

*"We are convinced that many things will sort themselves out if you just let things take their course and don't disturb [. . .], then usually it will go well."* (partner 4, UC)

Several men stated that they believed it was very important for the woman to feel good about the plans for the impending birth, since she was going to be the one who had to give birth:

*"What I can say about it, is that how [she] sees it and how [she] feels it, that that comes first. If [she] says now: '[next time] I want a C section up front', then I would find that very difficult, but eventually I would support that."* (partner 17, home breech perinatal death)

*"And then I felt like: if this is what you want, we'll do it. Then there is only one thing I can do, and that is get behind her decision and support her in it."* (partner 13, home VBAC)

Almost all partners reported that through conversation with their wives/girlfriends, they had become convinced by the women's arguments and had developed a shared vision on the nature of childbirth, in which an intimate, undisturbed home environment played a large role in the chance of a successful normal birth.

## Defending our views

Having established a shared vision, most partners broached the subject of going against medical advice with their family and friends. The reactions they encountered ranged from supportive to outright hostile. The most frequent response they received was an aversion to risk taking in childbirth, which led family members and friends to counsel against the couple's plans and

in favor of a birth within protocol. Even though they had become convinced of the merit of the intended birth plan, partners found it difficult to explain their reasoning in their own social circle, which sometimes left them feeling insecure about the impending birth:

> *"Their first reaction was very intense: 'We do not approve of this! Do you even know what you are doing? You are insane!' All sorts of phrases rained down on us. And yes, that made me insecure." (partner 15, attempted home breech birth)*

Some family members/friends considered midwives and obstetricians to be the experts and couples were warned that people could point fingers if things ended badly:

> *"I think for every lay person, you know, who did not grow up with this, who grew up with the idea of: it happens in the hospital, so there will be a need for professionals. . . I think that can quickly give rise to the idea that having an unassisted home birth, with nobody there, not even a midwife, that is scary, that is dangerous, etcetera. That is, I think, the first reaction you have." (partner 16, intended UC)*

Some partners did not discuss their plans with anyone at all, whereas others were actually inspired by other couples of their acquaintance who had gone down the same route. A few couples encountered favorable responses from their social circle, and even attempted to convince others to make the same birth plans, after their birth had gone well:

> *"And when you tell people how it went and [they] think: 'Oh, I might want that too, or not, and [they] think: ok.' Yes, it is quite the conversation starter, and I think that is fun and nice." (partner 19, UC)*

In addition to having to defend their plans in their own social circle, some partners also felt they had to defend the women against their care providers. In two cases, the partners even went to a hospital appointment alone, without the woman, to confront their obstetrician:

> *"The last week before the due date the doctor wanted another meeting. [Wife] felt like: 'I don't want to talk unless he has something [new] to offer.' [. . .] She didn't want to go, so I said: I will go and talk to him. [. . .] It was a pretty stressful meeting. I was glad [wife] wasn't there. She had gone through enough." (partner 8, home VBAC)*

In conclusion, after having established a shared vision, most partners took it upon themselves to defend the couple's plans in their own social circle, and some attempted to convince others, sometimes against negative responses. In addition, two partners even confronted a medical professional in order to defend their wife/girlfriend's wishes.

### Doing it together

Having developed a shared vision and discussed their intentions for the birth in their own social circle, the couples started preparing together for the birth they had decided on. Most of the partners were very supportive. Some reported taking a preparatory class like hypnobirthing together with their wives/girlfriends, or going with the women to a designated clinic for couples who are considering birth choices against medical advice, in order to get all the information available. Several couples discussed their intentions with an obstetrician at their local hospital, so there would be a record of the situation and their intentions in case they needed help during the birth:

*"We went to the hospital just so we had seen them and to say: 'Look, if things unexpectedly go differently, then we will come here, because we live five minutes away, so this is the hospital we will go to.'" (partner 17, home breech perinatal death)*

Another way to prepare for an unassisted childbirth was getting some selected tests, so the couple would not be surprised during the birth by, for example, twins:

*"With the fifth [child] we eventually had an ultrasound done. But that was more because there was so much movement inside her abdomen that we thought: OK, it looks like there are too many limbs there." (partner 5, UC)*

Some couples made extensive and well thought out plans together for what to do in case they came across certain complications during the birth, whereas others decided doing so would only become a self-fulfilling prophecy:

*"Yes, I asked those questions, like what if she suddenly starts bleeding, you would have to call the ambulance. [. . .] Or what if there is meconium in the water. Or what if the cord is around the head. [. . .] [During the birth] I listened to [wife] very carefully for signs that she might need help.[. . .] If she would start acting funny then there would be a problem, but she didn't." (partner 5, UC)*

*"But actually we did the same thing as when you get into a car. You don't think like: oh, if I get into an accident I will do such and such. We basically always assumed things would go well." (partner 4, UC)*

Most of the partners report writing an extensive birth plan together, often concerning intentions for a water birth. In a few cases, the unavailability of the option for a water birth in regular care was part of the process that led to the decision to choose home birth:

*"Well, the hospitals in this area didn't offer that [a water birth]. That was confirmed during the conversation [in the clinic]. If during that conversation it had turned out that 'Listen, that is possible here, with [midwife]', [. . .] you could do it in the bath, and. . .That didn't happen, so [. . .] we would have had to go elsewhere for that." (partner 17, home breech perinatal death)*

For some couples, the choice for having a home birth in a high risk pregnancy meant they had to find another midwife, since obstetricians do not attend home births, and their regular community midwives were unwilling to attend a high risk birth at home. Having to find a new provider, sometimes in the final stages of pregnancy, was a source of stress for the partners and their wives/girlfriends. The few midwives that were willing to attend such births were few and far between, meaning that travel distances could be further than couples would have liked:

*"That was quite a search. [wife] has telephoned a lot of people, and. . . look, because we were quite far along in the pregnancy, the time you have left is quite short. That may limit your options [. . .]." (partner 11, home VBAC)*

When a midwife was found, it was also considered important by partners that there would be a 'click', a connection with the midwife in question and that the wishes of the couple would come first, not the guidelines:

*"When I shook her hand and looked into her eyes, I thought: this is the one."* (partner 1, previous PPH and MPV)

*"Someone to just be there, and attuned to our needs. Instead of us having to accommodate her, so to speak."* (partner 19, UC)

In cases of intended unassisted childbirth, partners reported that they chose this option because the couple did not see a great deal of use for a midwife, other than just to be there in case problems arose during the birth:

*"To my mind, the best scenario would be if you could deliver unassisted with a midwife present who just sits on the corner of the sofa with a cup of tea, and does nothing unless she is asked."* (partner 4, UC)

Finally, partners spoke of what they believed their own role during the birth should be. Several partners stated that they felt the role of the partner during both pregnancy and birth should be much larger than it currently is:

*"I think there is a world to gain in how you can help your wife to get in the right mood, that she almost detaches a little from the world, feels safe, calm, and. . .that the process gets going and the birth goes smoothly. [. . .] Because right now, you are a bit like an appendage, you know. . .[. . .] You have your role, but you are not the one who will be doing it."* (partner 1, previous PPH and MPV)

The partners wanted to be involved and felt that they should be both constant companion and defender of the woman, and help her get through the birth from start to finish, doing it together:

*"Listen, you [as a partner] have a role during the birth. You need to make sure your wife is heard [. . .], and not overwhelmed by all the medical stuff."* (partner 5, UC)

*"I can be there for you [. . .]. I really want to be involved, and stay involved. And not that some protocol says I can't be there or I can't see certain things. I want to see them."* (partner 15, attempted home breech birth)

In summary, in addition to defending the couple's ideas to others, partners also took an active role in making detailed plans for the birth and finding a new midwife. They felt that they should also have a larger role during the birth itself, helping and supporting their wife in giving birth in the way that she wanted.

## Core theme: 'She convinced me'

After careful consideration of all four major themes and their subthemes, a pattern can be discerned, at the core of which is the process of the women convincing the partners of their views (Fig 3).

As stated previously, one partner brought up the idea of a home VBAC. In addition, two partners first suggested the idea of a water birth to the women, which may have been a factor in these couples' ultimate decision to go against medical advice, since water births are not always available in hospitals, or not for every high risk pregnancy. In all other cases, the women first suggested the idea for a home birth in a high risk pregnancy or a UC to the partners, did most of the research, and discussed what they had found. These discussions were in-

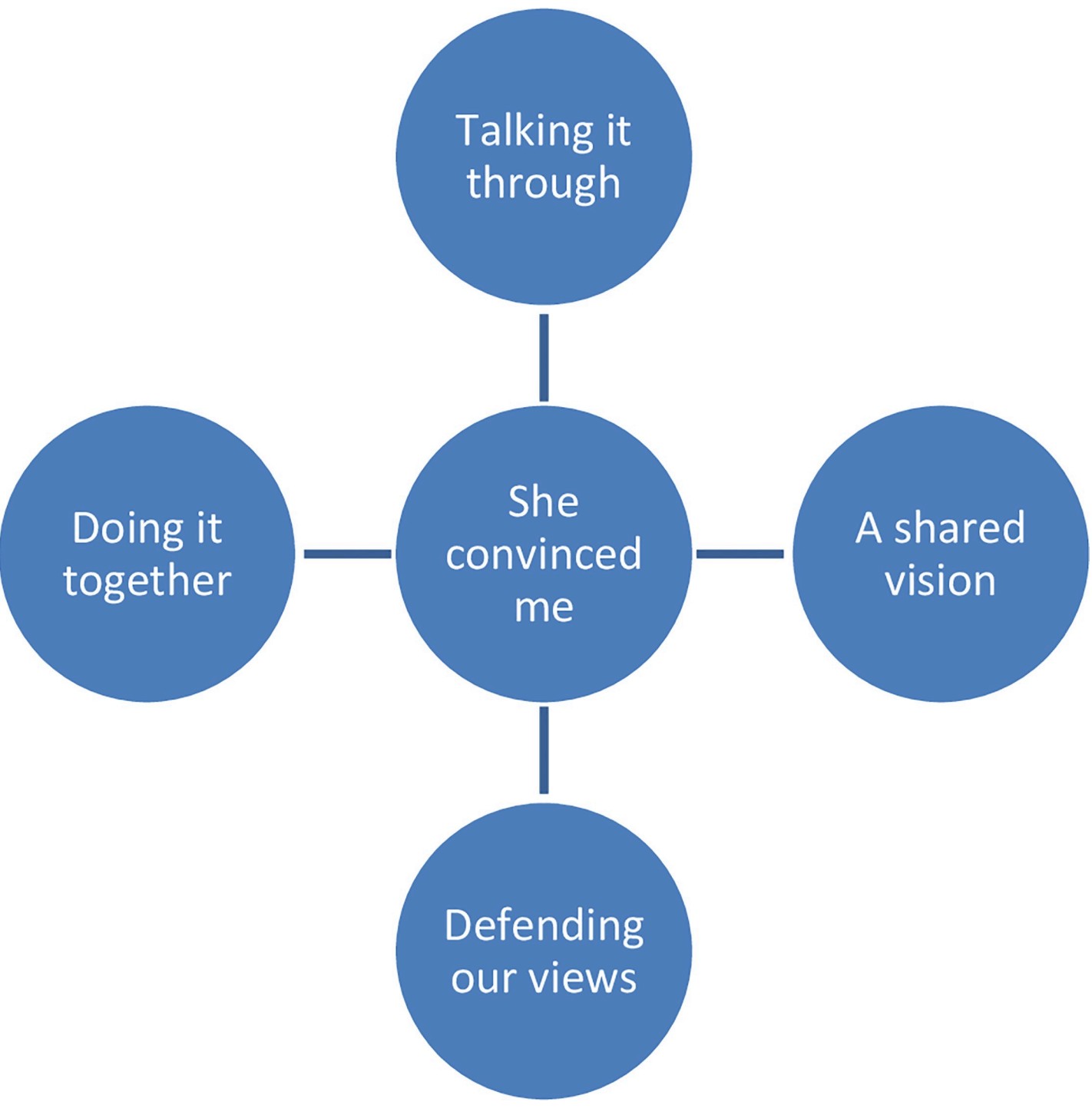

**Fig 3. Main themes diagram.**

depth, during which much soul-searching on the side of the partners took place. Through these discussions, the partners became convinced not only of the merit of the women's plans, but also of their objections to regular care and of their views on birth in general. This led to a shared vision for the desired care, and caused the partners to defend the birth plan to other

parties, as well as actively participate in (preparations for) the actual event. Preparing for and going through the process of these births was, for most partners, an intense experience, which strengthened the bond in their relationship, and made them come through this stronger as a couple:

*"The fact that she provided information for all my questions, either scientific articles or just other stories, you know. I thought: ok, well, this has been researched so thoroughly by her, that I thought, I can support this. [. . .] It sort of happened to me, you know. She started doing the research and I sort of got sucked into it." (partner 2, home VBAC)*

*"It was mostly my wife, I have to say. What she gave to me, I read, but she was the one who did all the research. [. . .] There is so much information on the internet. You can't just trust everything. [. . .] But when she dives into something, she does it right, you know? [. . .] But at first I was very critical. When I first heard of it [UC] I thought. . .I don't know about this. It can't hurt to have somebody there. But when we talked about it a lot, and [wife] had explained to me what the effect can be of just somebody being present. . ..then I understood what she meant and how it works. And then I was convinced in the end that we can do without [a midwife]" (partner 4, UC)*

## A negative case

Even in both cases where there was a bad outcome (perinatal death), the partners stated that, although things ended badly, they were still convinced the choice that was made was the right one, and they would make the same choice again if necessary. They reasoned that, when a decision is made on the right grounds, it is always the right decision, no matter how things end. However, one partner (partner 12, attempted twin home birth) differed from all other partners in respect that, in hindsight, he had some serious misgivings about the choices that were made prior to and during his wife's birth. She persuaded him to go along with her plan for home birth by showing him research she found online, which left him not entirely convinced:

*"At some point you think: fine, you do the research then. On the other hand, it did occur to me: aren't you using those articles that are the most convenient for you? [. . .] I didn't know that much about it, true. Then I would have to search for studies myself which would refute what [she] found. Then what would we actually be doing? That makes no sense at all." (partner 12, attempted twin home birth)*

His wife found a midwife on the other side of the country, which also struck him as strange:

*"I didn't actually realize that [our choices] were that unusual until [wife] said: 'I found a midwife from [across the country].' I thought: wow, what? If you have to go that far afield, [twin home birth] must be a strange step to take. [. . .] I did not actually talk to the midwife until the birth itself and. . .then I was not completely convinced and sure that this was the right decision." (partner 12, attempted twin home birth)*

At one point he was no longer fully on board with proceeding with a home birth, as he felt his family was threatened by social services being called in due to his wife's refusal to go to the hospital:

*"Until that point it was [wife's] birth, [. . .] but at that moment something changed, because suddenly this was about my family too. [. . .] I had reached the point where I thought: just give*

*in and go to the hospital already, because….you may have some wishes concerning your birth, but how much are you willing to risk to make that happen?" (partner 12, attempted twin home birth)*

Much later, the partner and his wife discussed the situation, but still did not agree:

*From what I saw of her here, I did not have a lot of confidence in [the midwife], I must say. […] It does not make conversations at home any easier. I have been told: 'You are just like those doctors.' Yes. It has been a pretty difficult time." (partner 12, attempted twin home birth)*

## Discussion

This qualitative study involved in depth interviews with 21 male partners of Dutch women who gave birth at home in a high risk pregnancy or had an unassisted childbirth, choices that are explicitly against medical advice.

### The women take the lead and filter information

Unlike in the study by Bedwell et al.[17], not all partners in this study automatically assumed that the birth would take place in a hospital, even in cases of a high risk pregnancy. In fact, most of them initially had no firm views about the impending birth and would not have raised the idea of going outside conventional care themselves. This is in line with other studies[18–20,22,33], where the pregnant woman initiated conversations about birth options that were against convention or medical recommendation. Often these plans originated in a previous traumatic experience in maternity care, where the women experienced a cascade of interventions and lack of shared decision making[18,20,24,28]. The overarching theme of 'fear' (of unnecessary intervention, loss of autonomy, provider's fear of bad outcome and legal ramifications) from our previous study on women's motivations[24] was not found to be an important factor in this current study. It appears that, in accordance with Ryding et al.[34], for the women, fear played a large role in decision making, whereas the men did not initially fear the regular system with its interventions, but mostly followed the women. In addition, the men did not seem to follow the women out of fear of upsetting them, or causing marital troubles, but out of a genuine wish to support them.

In contrast with other studies[4,18,19,21,35], where male partners did a substantial amount of research themselves, many partners in this study stated that the women extensively researched the risks and benefits of their preferred birth plan online and through social media, whereas the partners themselves read very little or nothing at all. They let their emotions, in this case their affection for their partner, guide their decision to agree to the proposed plan. Those partners that did do some research of their own mostly wanted to confirm what the women were telling them, since, as in our negative case, otherwise they would have to search for studies themselves which would refute what the woman found, and "That would make no sense at all". This phenomenon is known as 'confirmation bias', and is a well-known mechanism, that results in people believing they have independently confirmed the truth of what they have been told, when, in effect, they have not looked for anything disproving this information[36].

In addition, in accordance with the findings by Longworth[11] and Dejoy[33], several partners felt that it was 'her body, therefore her choice', and that they were happy with whatever the woman decided to do and would support her no matter what.

This shared vision was evident through remarkable similarities in statements about the regular maternity care system between the partners in this study and their wives/girlfriends in our previous study[24]. The interviews read like the women were a 'filter', through which the men acquired and interpreted information. This finding presents the ethical dilemma of care providers informing partners. Dutch law states that health care providers have an obligation to provide clear information regarding treatment, alternative treatments and preventative services to their patients, as long as they are competent to decide on matters pertaining to their own medical condition[37]. In the case of maternity services, this contract is with the woman. The medical professional is not required or even allowed to contact the partner to verify his understanding of the risks and benefits of the several options, or his opinion on the management plan. However, the woman's partner is presumably just as invested in the wellbeing of the child the woman is carrying as she is, and will be equally responsible for making decisions for the child as it's mother, after it is born. If the woman filters the information the partner receives, the partner may not be in possession of unbiased information from all possible sources and may therefore be unable to fully decide for himself how to weigh the risks and benefits of the proposed plan. However, it is not known how many of these partners were present at consultations with maternity care providers. It is possible that they did have the opportunity to be counseled by medical personnel, but simply attached more value to the arguments presented by their wife/girlfriend. In addition, even if partners had been present for most or all medical consultations, there is still no guarantee that medical personnel would have provided adequate counseling. Nonetheless, it seems possible that, as in our negative case, some partners of women who choose a high risk birth setting against medical advice may, in hindsight, be unhappy with decisions that were made. This could lead not only to marital conflicts, but also to remonstrations and dissatisfaction with health care professionals, if the partner feels that he would have made a different choice if he had been provided with all the existing evidence-based medical information.

## The partner as engaged protector

As stated in the introduction, several previous studies have shown that, contrary to the findings of this study, men are quite often not very involved in the process of pregnancy and childbirth[2–5]. Engaged partners have been shown to have a positive effect on their children and their wives/girlfriends: Jeffery et al[2]. found that increased levels of partner engagement in pregnancy and childbirth leads to better birth outcomes, and to a better bond between father and child. This is confirmed by Draper et al.[38], who state that engaged fathers are better fathers to their children. In addition, Xue et al.[14] in their study from Singapore, found that a higher level of engagement in fathers leads to lower incidences of postpartum depression in mothers, and to a better relationship between spouses.

The partners in the current study had a high level of engagement with the process of pregnancy and childbirth. Similar to studies by Jouhki et al.[20] and Viissainen[28], partners took on an active role participating in preparing for the birth, writing a birth plan, and discussing possible scenarios for which solutions were agreed on, making the men in this study appear to be very much engaged in the birth of their child.

Dejoy in her 2011 dissertation on the role of male partners in decision making around childbirth, found six main roles for partners in this process[33]: bystander, researcher, interpreter, leader/decider, limiter/boundary setter and protector. Only the role of protector seems to apply to the partners in this study.

Once they were convinced by their wife/girlfriend of the intended plan for the birth, they defended these views to their social circle. In accordance with the findings of Fenwick[3],

Sweeney[18], Jouhki[20] and Lindgren[21], they encountered mostly negative reactions from friends and family.

Two partners went even further in their role as protector of the woman, and went to the hospital in her place to confront a maternity care provider. This phenomenon can also be found in the studies by Locock[4] and Draper[38], who also describe the partner advocating for his wife/girlfriend against medical personnel. The fact that these partners felt that they needed to attend a hospital appointment with a medical professional in the woman's place, is indication of an irreversible breakdown in communication and cooperation between the professionals and the women involved. It illustrates how, in cases of substantially differing views between women and their caregivers, both parties can end up on opposing sides to such an extent that further cooperation becomes impossible. This situation then leads to a defining moment to go against medical advice, as illustrated in a multiple case study by Holten et al.[39] The burden is on the professionals to prevent the situation from escalating, by metaphorically and physically positioning themselves beside the patient, instead of on opposite sides.

### Implications for practice

This study shows, that in cases of birth wishes against recommendations, the women were the main and frequently the suspected only source of information for the men, and convinced the men of their ideas and plans. Therefore, when maternity care professionals are confronted with a pregnant woman who is planning on going against advice in her birth choices, it would be helpful to realize that she will most likely not make this decision alone. As Osamor and Grady[40] state, "Since 'others' will always be part of the exercise of one's agency in some form or other, interdependence should be recognized as the norm rather than independence (p.3)." It may thus be helpful to strive to discuss the options, risks and benefits of all possible scenarios in the presence of the partner. Doing so will remove the 'filter' the woman may put on all the information he receives, and help him in making up his own mind with all available data. This requires awareness of caregivers of their own risk perception and the willingness to provide bias free evidence based information. Counseling in this manner could prevent recriminations and dissatisfaction on the side of the partner after the event, since, as Osamor and Grady[40] suggest (p.5): "In certain contexts, a woman making decisions alone implies that [. . .] she is in fact bearing the burden of full responsibility and potential blame for those decisions."

There is currently a lack of (internet) sources of information specifically designed for partners of pregnant women, with almost all sources focusing on the women themselves. We therefore recommend the construction of specific websites for and by prospective fathers and care providers, which not only provide information but also focus on decision making.

Finally, a situation in which the chasm between professionals and pregnant women becomes so wide that it becomes necessary for the partner to advocate for the woman is an undesirable situation. Health care professionals should never allow themselves to end up on opposite sides from their patient, even if there is a disagreement on the best management plan. Instead, they should position themselves beside the woman, both physically and metaphorically. They should practice shared decision making[41], explore all options with the couple, using actual percentages rather than odds ratios, and discussing numbers needed to treat and numbers needed to harm. The aim of counseling should not be to frighten the couple into submitting to recommended treatment, but to fully inform. Furthermore, professionals should be willing to make compromises and allow for second best care in order to prevent, from the professionals' point of view, undesirable choices as the ones described in the article.

## Strengths and limitations

There are several strengths to this study. First, for a qualitative study, it is large, with 21 in-depth interviews with men with varying levels of education and different ages. Second, it is the first study on partner involvement in decisions regarding birth choices against medical advice in the Netherlands. This is important because it is likely that these occurrences are relatively rare here, since home birth for low risk women is integrated in the regular maternity care system, and therefore not against medical advice. Third, it is part of a larger project[27], for which the wives/girlfriends of the men were also interviewed. This has helped us to triangulate the results of both studies, and has heightened validity. Last, data saturation was reached on not only the main themes, but also the subthemes.

Naturally, there are also several limitations to this study. First, in approximately a third of cases the wife/girlfriend was present during the interview, which may have caused partners to give certain desirable answers. However, we did not discern any noticeable differences in answers between partners who were interviewed alone, and those who had their wife/girlfriend present. Second, we did not interview any partners whose wife/girlfriend wanted to go against medical advice in her birth choices, but where the partner convinced the woman to stay within regular maternity care. This is an important point, since it is likely that there may be many more of those cases, but they are impossible to trace, and therefore, no comparison on partner involvement in decision making can be made. Third, there was no member check in the form of either returning transcripts to the participants or organizing a feedback focus group. This was deemed impossible and/or impractical due to logistic and time restraints. However, we did not rely on field notes alone, rather, all interviews were transcribed verbatim. Therefore, there is little doubt concerning the actual words used by the participants. Fourth, since the partners were interviewed months to (in a few cases) several years after the events leading up to the relevant births, it is possible that, with the benefit of hindsight, they appear to have given the considerations concerning these births more thought and meaning than they actually did at the time. Finally, the interviews were performed by researchers from the field, with extensive background in Dutch maternity care, and a professional interest in birth choices against medical advice. It is possible that this fact influenced the tone and content of the interviews. Perhaps some participants were more, or less, negative about Dutch maternity care than they would otherwise have been. On the other hand, their professional expertise allowed the researchers to formulate pertinent questions, which helped in constructing a relevant topic list and likely added depth to the interviews.

## Conclusion

This qualitative interview study examines the involvement of partners in decision making concerning choices for a high risk birth setting against medical advice in the Netherlands. Four main themes were found: 1) Talking it through, 2) A shared vision, 3) Defending our views, and 4) Doing it together. From the data, one overarching theme emerged, and that was "She convinced me". This study shows that the idea for having a home birth in a high risk pregnancy or an unassisted childbirth almost always originates with the women, who seem to be the main source of information for the partners. This information appears to be 'filtered' by the women, and convinces the partners of the merit of the women's plans. They adopt and assimilate the women's views on childbirth. They support these views, which are now shared, by defending the plan for a birth against medical advice in their social circle, as well as in contacts with maternity care providers. Once convinced, the partners are very involved in planning and preparing for the birth. Maternity care providers can use these data to attempt to involve partners more during consultations in pregnancy, especially in cases where there is a

discrepancy between the wishes of the woman and the advice of the professional. That will ensure that partners also receive information on all options, risks and benefits of possible birth choices, and that they are truly in support of a final plan. More research is needed into partners' satisfaction with births both inside and outside the system.

## Supporting information

**S1 File. Keywords and list of abbreviations.**
(DOCX)

## Acknowledgments

The authors wish to thank the 21 men who took time out of their busy lives to share with us their thoughts and experiences on giving birth in a setting that went against medical advice.

We would also like to thank the following medical students for transcribing many hours of interviews: Sabine de Wild, Esther Bouwer, Vreni Bron, Cynthia ter Horst, Roelie de Jager, Jorinde Westra, and Julia van Ling.

Last but not least we would like to thank the designated outpatient clinic of the Amsterdam UMC, Amsterdam, the Netherlands, for helpful suggestions of potential participants.

## Author Contributions

**Conceptualization:** Martine Hollander, Esteriek de Miranda, Frank Vandenbussche, Jeroen van Dillen, Lianne Holten.

**Data curation:** Martine Hollander, Lianne Holten.

**Formal analysis:** Martine Hollander, Lianne Holten.

**Funding acquisition:** Esteriek de Miranda, Frank Vandenbussche, Lianne Holten.

**Investigation:** Martine Hollander, Esteriek de Miranda, Lianne Holten.

**Methodology:** Martine Hollander, Lianne Holten.

**Project administration:** Esteriek de Miranda.

**Supervision:** Esteriek de Miranda, Frank Vandenbussche, Jeroen van Dillen, Lianne Holten.

**Writing – original draft:** Martine Hollander, Lianne Holten.

**Writing – review & editing:** Martine Hollander, Esteriek de Miranda, Anne-Marike Smit, Irene de Graaf, Frank Vandenbussche, Jeroen van Dillen, Lianne Holten.

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
