## [Decision Letter · Decision Letter 0]

18 Nov 2019

PONE-D-19-18382

‘She convinced me’- Partner involvement in choosing a high risk birth setting against medical advice in the Netherlands: a qualitative analysis.

PLOS ONE

Dear Dr. Hollander,

Thank you for submitting your manuscript to PLOS ONE. After careful consideration, we feel that it has merit but does not fully meet PLOS ONE’s publication criteria as it currently stands. Therefore, we invite you to submit a revised version of the manuscript that addresses the points raised during the review process.

Please note that Reviewer 3 recommended a "reject decision". We will thus consider absolutely necessary that you address in your revised manuscript all the methodological points raised by this reviewer, including the number of interviewed participants and the statistical analysis. We are aware that given that supplemental investigations  recommended, you may decide not to revise and would perfectly understand such decision.       

We would appreciate receiving your revised manuscript by Dec 26 2019 11:59PM. To enhance the reproducibility of your results, we recommend that if applicable you deposit your laboratory protocols in protocols.io, where a protocol can be assigned its own identifier (DOI) such that it can be cited independently in the future. For instructions see: http://journals.plos.org/plosone/s/submission-guidelines#loc-laboratory-protocols

We look forward to receiving your revised manuscript.

Kind regards,

Umberto Simeoni

Academic Editor

PLOS ONE

Journal Requirements:

1. Please include captions for your Supporting Information files at the end of your manuscript, and update any in-text citations to match accordingly. Please see our Supporting Information guidelines for more information: http://journals.plos.org/plosone/s/supporting-information.

2. Please upload a new copy of Figure 1 as the detail is not clear. Please follow the link for more information: http://blogs.PLOS.org/everyone/2011/05/10/how-to-check-your-manuscript-image-quality-in-editorial-manager/

Reviewers' comments:

Reviewer's Responses to Questions

**Comments to the Author**

1. Is the manuscript technically sound, and do the data support the conclusions?

Reviewer #1: Yes

Reviewer #2: Yes

Reviewer #3: No

2. Has the statistical analysis been performed appropriately and rigorously? 

Reviewer #1: N/A

Reviewer #2: N/A

Reviewer #3: No

3. Have the authors made all data underlying the findings in their manuscript fully available?

Reviewer #1: No

Reviewer #2: No

Reviewer #3: Yes

4. Is the manuscript presented in an intelligible fashion and written in standard English?

Reviewer #1: Yes

Reviewer #2: Yes

Reviewer #3: Yes

5. Review Comments to the Author

Reviewer #1: This qualitative study explores the involvement of 21 partners in the dmp regarding a home birth in a high risk pregnancy or unassisted childbirth.

The study design is adapted. The researchers might specify whether the research paradigm is to aim for change in care practices or whether a specific objective for the organisation of care was targeted.

It would be necessary to specify at the level of methodology the a priori of the researchers who are mentioned in an implicit way in the discussion.

Ethical obligations have been met.

Results are clear and well presented. Themes and sub themes are illustrated with verbatims.

Regarding the motivations of the fathers, it seems that they mainly followed their wives. Were the researchers able to assess whether fear was a driving force among the partners, in terms of fear, whether it was to generate a couple conflict, or to upset their pregnant wife ?

Finally, was the (future) child's well-being present in their speech ? If not, this could be mentioned.

Some remarks regarding the discussion

The authors might avoid some repetitions in the discussion and conclusion.

They could broaden the point of view on this dm issue by mentioning the role of emotions as the driving force behind decisions and the possible impact of cognitive biases, especially the "confirmation bias" that can be imagined in the way information transmitted to partners is described by some partners as "sorted" and accepted quite passively by them.(see Joyce Ehrlinger et al Decision-making and cognitive biases dec 2016 DOI: 10.1016/B978-0-12-397045-9.00206-8

In the limitations section, it might be useful to mention that these results are words that come after the facts, and can therefore retroactively inject a more reflective and constructed meaning into initially mainly intuitive attitudes.

The practical implications are very important. The authors could add a comment/proposal in terms of Internet information resources constructed together by patient support groups and professionals. A table could be added with proposals to better integrate fathers in the dmp. (?)

Globally, the research paper is valuable:

this work can concretely promote decisions involving partners. This work is original, and deals with a public health problem with significant health implications.

This work can concretely promote decisions involving partners, who are also future fathers.

Reviewer #2: I have read with great interest the paper of M. Hollander et al. and I think it is of scientific quality high enough and deserves publication in PLOS, providing the authors are willing to give some precisions.

In summary, the authors conducted a qualitative research in order to understand better the role of women’ partners choosing high risk birth setting against medical advice. They interviewed 21 subjects and found 4 main themes: “Talking it through”, “A shared vision”, “Defending our views”, “Doing it together”. The overarching theme that covered all the others was “she convinced me”.

First, I would like to congratulate the authors for their hard work putting through so many interviews and for the quality of their transcriptions.

In general, the manuscript is well written and easy to read. However, it is a little bit long and the authors should go forward in their discussion and not only rephrase most of the results.

A weakness of this study also lies in its retrospective nature and sometimes the long delay between pregnancy / delivery and the interview. This could have introduced a recognition bias. Furthermore, can the authors precise if the couples had subsequent pregnancies and their choices for delivery / issue of the pregnancy?

The fact that the authors interviewed the partners of women previously interviewed in another study could also have introduced a bias as they might have read the results of the previous study. Obviously, this is also a strength of this study since it helped increasing the number of partners interviewed.

Theses limits should appear clearly in their discussion.

By any chance, did the authors observed if the partner answers were different when the woman was / was not here? Did they have a chance to transcript some of the physical language not otherwise expressed?

When reading the study, I can’t help wonder whether the way these couple are dealing with the choice of delivery is unique or are they reproducing a pattern they are used to for different fields? Like buying a house, choosing where to live, work, etc etc….

It would have been great to submit a general questionnaire to the couples to understand the way they usually process decisions and see whether their choice for birth place is different or not.

Reviewer #3: This is a small study of fathers role in choosing home births in high risk patients.

-21 subjects is a very small study and may not have statistical validity

You should ty to collect at least 50 subjects.

You state that “This study is part of a larger project exploring out of the system birth, the

Wonderstudy27, in which we also interviewed women who went against medical advice in

their birth choices (home birth in a high risk pregnancy or UC), their partners and their

midwives” therefore you should easily be able to recruit the required additional subjects.

Line 146 you state “and two partners were interviewed twice.” Please explain why

You state that “Most often the partner was interviewed alone, although in several cases, the woman was present in the room and occasionally joined in the conversation.” Please discuss how this may have biased your results.

“Interviews were semi-structured by use of a topic list (Figure 1)” is this a validated question list. Please state yes or no. if no justify why it could be used.

Did you consider doing any statistics on your finding

Otherwise it has no support for any statement you use.

Please list indications that the birth was transferred to the hospital, in those cases

What special questions were prepared relative to the two intra-labor demises.

I do not believe I can make any valid conclusion from the manner the data was collected and with the lack of statistical analysis. I will stop here.

6. PLOS authors have the option to publish the peer review history of their article (what does this mean?). If published, this will include your full peer review and any attached files.

Reviewer #1: Yes: Laurence Caeymaex

Reviewer #2: Yes: Yohann Dabi

Reviewer #3: Yes: Michael H. dahan

---

## [Author Response · Author response to Decision Letter 0]

23 Dec 2019

1. Please include captions for your Supporting Information files at the end of your manuscript, and update any in-text citations to match accordingly. Please see our Supporting Information guidelines for more information: http://journals.plos.org/plosone/s/supporting-information.

We have included the captions at the end of the manuscript.

2. Please upload a new copy of Figure 1 as the detail is not clear. Please follow the link for more information: http://blogs.PLOS.org/everyone/2011/05/10/how-to-check-your-manuscript-image-quality-in-editorial-manager/

Unfortunately the video in this link does not play.

We have reconverted our .docx file to .tiff. We think the detail is now clear.

Reviewer #1: This qualitative study explores the involvement of 21 partners in the dmp regarding a home birth in a high risk pregnancy or unassisted childbirth.

The study design is adapted. The researchers might specify whether the research paradigm is to aim for change in care practices or whether a specific objective for the organisation of care was targeted.

We have added a statement to this effect at the end of the introduction.

It would be necessary to specify at the level of methodology the a priori of the researchers who are mentioned in an implicit way in the discussion.

We have expanded on the a priori position of the researchers in the method section of researchers and reflexivity.

Ethical obligations have been met.

Results are clear and well presented. Themes and sub themes are illustrated with verbatims.

Thank you.

Regarding the motivations of the fathers, it seems that they mainly followed their wives. Were the researchers able to assess whether fear was a driving force among the partners, in terms of fear, whether it was to generate a couple conflict, or to upset their pregnant wife ?

Good question. No, that seemed not to be the case. The word fear was hardly ever mentioned by the fathers. The partners were mostly very protective and supportive of the women. We have added a few words in the discussion.

Finally, was the (future) child's well-being present in their speech ? If not, this could be mentioned.

Yes, it was, in quite a few cases. We also mention this in the results, in quotes from partner 6 and partner 18, and in the discussion.

Some remarks regarding the discussion

The authors might avoid some repetitions in the discussion and conclusion.

We have removed quite a few sentences from the discussion. We understand that the conclusion can also read as repetitive, however, we feel it must report all the main findings, for those readers who choose not to read the entire article. Therefore we have left this segment intact.

They could broaden the point of view on this dm issue by mentioning the role of emotions as the driving force behind decisions and the possible impact of cognitive biases, especially the "confirmation bias" that can be imagined in the way information transmitted to partners is described by some partners as "sorted" and accepted quite passively by them.(see Joyce Ehrlinger et al Decision-making and cognitive biases dec 2016 DOI: 10.1016/B978-0-12-397045-9.00206-8

We agree with this thought and have included several sentences to this effect in the discussion.

In the limitations section, it might be useful to mention that these results are words that come after the facts, and can therefore retroactively inject a more reflective and constructed meaning into initially mainly intuitive attitudes.

We have added a few words to this effect.

The practical implications are very important. The authors could add a comment/proposal in terms of Internet information resources constructed together by patient support groups and professionals. A table could be added with proposals to better integrate fathers in the dmp. (?)

A comment on internet sources has been added.

A table with proposals for integrating fathers is a useful but perhaps ambitious suggestion. There are already quite a few tables in this article, therefore we have elected not to create yet another. That could, however, be the focus of another (our next?) study on partner involvement.

Globally, the research paper is valuable:

this work can concretely promote decisions involving partners. This work is original, and deals with a public health problem with significant health implications.

This work can concretely promote decisions involving partners, who are also future fathers.

Thank you!

Reviewer #2: I have read with great interest the paper of M. Hollander et al. and I think it is of scientific quality high enough and deserves publication in PLOS, providing the authors are willing to give some precisions.

Thank you, and of course!

In summary, the authors conducted a qualitative research in order to understand better the role of women’ partners choosing high risk birth setting against medical advice. They interviewed 21 subjects and found 4 main themes: “Talking it through”, “A shared vision”, “Defending our views”, “Doing it together”. The overarching theme that covered all the others was “she convinced me”.

First, I would like to congratulate the authors for their hard work putting through so many interviews and for the quality of their transcriptions.

Thank you again!

In general, the manuscript is well written and easy to read. However, it is a little bit long and the authors should go forward in their discussion and not only rephrase most of the results.

We have already done this in response to reviewer 1’s comments. The discussion is now less repetitive, and shorter than the one in our last PLoS One publication regarding the midwives involved in these cases (PLoS One. 2019 Jul 30;14(7):e0220489. doi: 10.1371/journal.pone.0220489).

A weakness of this study also lies in its retrospective nature and sometimes the long delay between pregnancy / delivery and the interview. This could have introduced a recognition bias. Furthermore, can the authors precise if the couples had subsequent pregnancies and their choices for delivery / issue of the pregnancy?

We have commented on the time lapse in the limitations section in response to reviewer 1.

As for any subsequent pregnancies which may have occurred since we did the interviews in 2014-2016, we are aware of a few, through informal channels, although we have not formally spoken to the couples about this. With most couples there has been no further contact. Those subsequent births that we know of have all been against medical advice, once again.

Since this is informal information on only a few cases we are not including this in the article.

The fact that the authors interviewed the partners of women previously interviewed in another study could also have introduced a bias as they might have read the results of the previous study. Obviously, this is also a strength of this study since it helped increasing the number of partners interviewed.

Theses limits should appear clearly in their discussion.

Actually, the interviews with the partners took place around the same time (and often consecutively) as those with the women. The article on the women’s interviews was published approximately one year after we interviewed the last partner, so at the time of these interviews they were unaware of those results (as were we).

By any chance, did the authors observed if the partner answers were different when the woman was / was not here? Did they have a chance to transcript some of the physical language not otherwise expressed?

To the best of my recollection (and I did the vast majority of these interviews) there were no notable differences. We have added a statement to that effect. I also did not notice any body language conveying any pressure on the partner to give certain desirable answers. The couples appeared very much ‘in sync’.

When reading the study, I can’t help wonder whether the way these couple are dealing with the choice of delivery is unique or are they reproducing a pattern they are used to for different fields? Like buying a house, choosing where to live, work, etc etc….

It would have been great to submit a general questionnaire to the couples to understand the way they usually process decisions and see whether their choice for birth place is different or not.

Indeed it would have. And I believe it might have shown the same pattern. But since we did not discuss their general decision making process, we do not know.

Thank you both very much for your insightful suggestions.

---

## [Decision Letter · Decision Letter 1]

30 Jan 2020

‘She convinced me’- Partner involvement in choosing a high risk birth setting against medical advice in the Netherlands: a qualitative analysis.

PONE-D-19-18382R1

Dear Dr. Hollander,

We are pleased to inform you that your manuscript has been judged scientifically suitable for publication and will be formally accepted for publication once it complies with all outstanding technical requirements.

Please note that this decision has been made based on the current criteria that prevail for qualitative research, after careful consideration of the criticisms made by one reviewer, who recommended to reject the manuscript.  

With kind regards,

Umberto Simeoni

Academic Editor

PLOS ONE

In the Methods, please clarify whether the participants gave written or verbal informed consent for their quotes to be used in this article.

Additional Editor Comments (optional):

Reviewers' comments:

Reviewer's Responses to Questions

**Comments to the Author**

1. If the authors have adequately addressed your comments raised in a previous round of review and you feel that this manuscript is now acceptable for publication, you may indicate that here to bypass the “Comments to the Author” section, enter your conflict of interest statement in the “Confidential to Editor” section, and submit your "Accept" recommendation.

Reviewer #1: All comments have been addressed

Reviewer #2: All comments have been addressed

2. Is the manuscript technically sound, and do the data support the conclusions?

Reviewer #1: Yes

Reviewer #2: Yes

3. Has the statistical analysis been performed appropriately and rigorously? 

Reviewer #1: N/A

Reviewer #2: N/A

4. Have the authors made all data underlying the findings in their manuscript fully available?

Reviewer #1: Yes

Reviewer #2: Yes

5. Is the manuscript presented in an intelligible fashion and written in standard English?

Reviewer #1: Yes

Reviewer #2: Yes

6. Review Comments to the Author

Reviewer #1: (No Response)

Reviewer #2: Thank you for revising your manuscript. While it remains quite long to allow a quick reading, the main comments I had (which were mostly shared with reviewer number 1) were adressed in the revised version.

Next time, think of giving couple a prior questionnary to understand the way they usually process life important decisions.

7. PLOS authors have the option to publish the peer review history of their article (what does this mean?). If published, this will include your full peer review and any attached files.

Reviewer #1: Yes: Laurence Caeymaex, MD, PhD

Reviewer #2: Yes: Yohann Dabi

---

## [Editor Report · Acceptance letter]

5 Feb 2020

PONE-D-19-18382R1 

‘She convinced me’- Partner involvement in choosing a high risk birth setting against medical advice in the Netherlands: a qualitative analysis. 

Dear Dr. Hollander:

I am pleased to inform you that your manuscript has been deemed suitable for publication in PLOS ONE. Congratulations! Your manuscript is now with our production department. 

With kind regards,

on behalf of

Dr. Umberto Simeoni 

Academic Editor

PLOS ONE